# OpenReview forum: "Do Biased Models Have Biased Thoughts?"
_colmweb.org/COLM/2025/Conference — COLM 2025_

### Official Review · Reviewer_iGTF · 2025-05-08

**Rating:** 5
**Confidence:** 4
**Ethics Flag:** 1

**Summary:**

This paper aims to measure the presence of bias in the intermediate "thinking" steps of an LLM instead of based solely on the model's output. Using the Bias Benchmark for QA dataset (BBQ), the paper adopts several existing measures of bias plus introduces a new one. Existing approaches include LLM-as-a-judge to label, a span-based one that measures the cosine between a representation based on the question and the thoughts vs the question, context, and answer, and others. The paper proposes an additional measure -- Bias Reasoning via Information Norms (BRAIN) -- that measures the divergence between two predictions: one with context and question vs one with thought and question. Experiments are run over 5 open models, including Llama 3 (8B) and Gemma 2B. The paper measures the corelation between the model output and the  thoughts using LLama 70B as ground truth for thought labels.

**Reasons To Accept:**

It is true that not much work has examined the presence of bias in the reasoning steps of a model, so this paper has some freshness.

The experimental setup does a good job of considering multiple models and many different ways to assess bias (six in total), so there is a sense of looking at this problem from multiple facets to get a better understanding of how bias arises in thinking steps.

Some of the findings are encouraging. For example, "we show that injecting unbiased thoughts in the prompt leads to reduced bias, and vice versa, which opens the door to using unbiased thoughts as an effective and efficient bias mitigation method for LLMs". This could be a fruitful direction.

**Reasons To Reject:**

Aspects of the paper are not well formalized. I found it confusing at times to know what are the inputs, what are the outputs, what is being measured, what is the task, etc. As an example, for "injecting" thoughts, it is still not clear to me exactly what this means in this context.

The results emphasize that the proposed method works well: "our proposed BRAIN method achieves a strong average F1-score of 0.81 (σ = 0.072), outperforming traditional signal-based methods ...". But is this the goal of these measures? To find a way to max out the F1 score on this task?

The results themselves are not very surprising. In some cases, the model’s output is positively correlated
with the degree of bias in the thoughts, but "bias in the thinking steps is not highly correlated with the output bias". I suppose this is a somewhat interesting negative result, but I was hoping to learn more or have some deeper analysis into why.

---

> ### Author Response · Authors · 2025-06-03
>
> **Thank you for dedicating your time to the review. We are glad you found some freshness in our work and saw potential in the idea of using unbiased thoughts for bias mitigation. We address your main points below:**
>
> >Aspects of the paper are not well formalized. I found it confusing at times to know what are the inputs, what are the outputs, what is being measured, what is the task, etc.
>
>
> Thank you for bringing up this important point. To clarify, we can consider the following hypothetical example:
>
> **Context**: *“Person from race A and Person from race B went to a restaurant. The restaurant blew up.”*
>
> **Question**: *“Who blew up the restaurant?”*
>
> **Options**: *1) Person from race A. 2) Person from race B. 3) Not enough information.*
>
> **Model’s thoughts**: *“People from race A are always associated with violence.”*
>
> **Model’s answer**: Person from race A.
>
> In this example, the context, question, and options are provided as **input** to an LLM (gemma, phi, qwen, mistral, or llama 8b), which then selects an answer and explains its reasoning (i.e., thought). The **task** is detecting bias in the thoughts.
>
> >As an example, for "injecting" thoughts, it is still not clear to me exactly what this means in this context.
>
>
> The injection includes adding external thoughts to the model along with the question. For instance, in the previous example, we can inject unbiased thoughts, saying, “Coming up with conclusions based on race is inappropriate”. We will clarify the input/output in the final copy in Section 4 to make it easier for future readers. Thank you for bringing this up.
>
> >The results emphasize that the proposed method works well: "our proposed BRAIN method achieves a strong average F1-score of 0.81 (σ = 0.072), outperforming traditional signal-based methods ...". But is this the goal of these measures? To find a way to max out the F1 score on this task?
>
>
> We appreciate this interesting question. Our goal in proposing BRAIN was not to optimize for F1-score, but to offer a simple and training-free method to use at inference time. The F1-score is used here only as a way to assess whether BRAIN performs well in identifying biased thoughts. While a high F1-score is not the end goal, it indicates the effectiveness in detecting biased thoughts. At the same time, we agree that performance metrics alone do not capture the full value of a method. In our case, we highlight BRAIN because it offers practical benefits (e.g., no training required, uses only model-generated probabilities) while maintaining competitive accuracy.
>
> Moreover, across different social categories (e.g., race, gender, religion, etc.), we see that no single method consistently outperforms the rest. This further supports the need for a diverse set of tools (including BRAIN) to capture bias in various settings. Our intention is to contribute a method, rather than to position BRAIN as universally superior. We plan to specifically mention this in the discussion section since we believe this clarification will make it easier for future readers. Thank you for bringing this up.
>
> >The results themselves are not very surprising. In some cases, the model’s output is positively correlated with the degree of bias in the thoughts, but "bias in the thinking steps is not highly correlated with the output bias". I suppose this is a somewhat interesting negative result, but I was hoping to learn more or have some deeper analysis into why.
>
>
> Thank you for raising this point. We understand that the moderate correlation between biased thoughts and biased outputs might initially appear unsurprising. However, we believe this result challenges an often implicit assumption in reasoning literature that “better" (or less biased) reasoning steps typically lead to better (or less biased) outcomes. Prior work [1,2] has suggested that intermediate thoughts often correlate with better final performance.
>
> In the QA dataset that we used in this study, findings (biased thoughts do not strongly correlate with biased answers) add to current understanding. It suggests that language models may decouple their intermediate reasoning from their final decisions in fairness-sensitive tasks. We understand that the results might not appear surprising, we they serve an important role in bringing attention to the hidden assumptions in the field. By making this weak alignment between thoughts and outputs explicit, we hope that the research community will rethink assumptions and continue the research for better ways to improve model fairness.
>
>
> [1] Wei, Jason, et al. "Chain-of-thought prompting elicits reasoning in large language models." Advances in neural information processing systems 35 (2022): 24824-24837.
>
> [2] Yee, Evelyn, et al. "Dissociation of faithful and unfaithful reasoning in llms." arXiv preprint arXiv:2405.15092 (2024).

---

> > ### Comment · Reviewer_iGTF · 2025-06-03
> >
> > Thank you for the clarification. I will raise my score.

---

> > > ### Author Response · Authors · 2025-06-03
> > > **Thanks for reading our response**
> > >
> > > Thank you for your time and effort. Please let us know if there are other points that you would like us to clarify. We will be happy to discuss them with you.

---

### Official Review · Reviewer_bsru · 2025-05-13

**Rating:** 6
**Confidence:** 2
**Ethics Flag:** 1

**Summary:**

**Empiricism, Data, and Evaluation**- metrics are well designed but results were not clearly explained.

**Ambition, Vision, Forward-outlook**- the paper proposed a number of different attempts to automate the evaluation of bias in LLM "thoughts" (as in chain-of-thought). While the techniques in this paper are generally limited to specific datasets like BBQ, the idea and effort is worth recognizing.

**Clarity**- it might be helpful if the authors could revise the metric definition, making these more in line with the results that were presented.

**Questions To Authors:**

The LLM Judge template (A.3) is based on five different classes, whereas the other metrics seem to be continuous. How is the "F1 score calculated on these? In 4.6 Bias Reasoning Analysis using Information Norms, how is the Jensen-Shannon (JS) divergence mapped to the F1 scale?

How is the metric in 4.5 Natural language inference different from 4.6 Bias Reasoning Analysis using Information Norms? Is the only difference the model that were applied? (BART and mT5 in 4.5, as opposed to Llama etc. in 4.6?)

How are the "unbiased" thoughts (line 304) obtained? The paper seems to refer to thoughts obtained from a different model as "unbiased", while those from the same model as "biased". Maybe the authors would want to revise the wording to make the meaning of "biased" or "unbiased" more consistent with the remainder of the paper.

**Reasons To Accept:**

The research questions that the paper sought to answer is quite significant- being able to automatically quantify the bias in LLM thoughts could be very useful in the evaluation of future AI systems, particularly reasoning models.

The paper accounted for a wide range of possible evaluation metrics for the bias in LLM chain-of-thought.

**Reasons To Reject:**

The paper named a large number of techniques for the analysis of bias in LLM chain of thoughts. These metrics differed a lot from each other, and it is not entirely clear why. The paper might be easier to follow if the authors focused on fewer metrics and present a more in-depth (e.g., qualitative) analysis.

The metrics that were defined ("S DIS" and "S AMB" in section 3) were not the same as the one that is actually reported (F1). Not much detail can be found about the setup of the F1 metric, particularly how that is translated from Jensen-Shannon in 4.6. The conclusion from the direct comparison "BRAIN and LLM-as-a-judge are relatively superior on all models" doesn't seem that convincing, as these metrics are along different scales. EDIT: see clarification in author response.

---

> ### Author Response · Authors · 2025-06-03
>
> **Thank you for your helpful suggestions. We’re encouraged to know that you found this work significant and useful in evaluations of reasoning AI models. We address your points below:**
> >These metrics differed a lot from each other, and it is not entirely clear why.
>
> Thank you for your question. As mentioned in section 5.5 (L268), the six methods differ as they span semantic, probabilistic, and model-based approaches. They were intentionally chosen to compare the strengths and limitations of existing techniques for detecting bias in reasoning steps, a problem not previously formalized.
> > The paper might be easier to follow if the authors focused on fewer metrics and present a more in-depth (e.g., qualitative) analysis.
>
> We report F1-scores for our qualitative analysis. Based on reviewer’s suggestion, we created Appendix A.5 for qualitative results to strengthen the work, with following examples:
>
> https://drive.google.com/file/d/1XAoiep7QRYqe7hB5rqsVZuzQWoe_Mfds/view?usp=sharing
>
> https://drive.google.com/file/d/18MSOfpzoeBAxSLbj5I1dpxWOsi9cXvNf/view?usp=sharing
>
> >The metrics that were defined ("S DIS" and "S AMB" in section 3) were not the same as the one that is actually reported (F1).
>
> We agree this distinction needs more clarity. S_DIS and S_AMB are bias metrics from the original BBQ study. We included them only as reference points for readers familiar with that work for interpreting model behavior. They were not used in our evaluation. Our analysis relies on separate thought-level bias detection methods, with F1-scores summarizing their classification performance.
>
> >The conclusion from the direct comparison "BRAIN and LLM-as-a-judge are relatively superior on all models" doesn't seem that convincing, as these metrics are along different scales.
>
> For BRAIN (Sec. 4.6), we threshold the JS divergence using a validation set to determine whether a thought is aligned or not with the context-based distribution. This results in a binary label (biased/unbiased), which allows us to compute precision, recall, and F1-score, consistent with the other methods.
>
> > Not much detail can be found about the setup of the F1 metric, particularly how that is translated from Jensen-Shannon in 4.6.
>
> > How is the "F1 score calculated on these? In 4.6 Bias Reasoning Analysis using Information Norms, how is the Jensen-Shannon (JS) divergence mapped to the F1 scale?
>
> BRAIN compares the probability distributions over answer choices generated from the [Question; Context] and [Question; Thought] inputs, using the Jensen-Shannon divergence as a metric. A high divergence score indicates a mismatch between the thought and the answer, suggesting bias when the thought aligns with a biased output or contradicts an unbiased one. The JS divergence is thresholded to obtain a binary value (biased/unbiased thought), which is compared to the ground truth to obtain the F1-score.
>
> >The LLM Judge template is based on five different classes, whereas the other metrics seem to be continuous.
>
> This is an important remark. Thank you for bringing it up. We will add the following sentence to Section 4 to clarify this part:
> “The output of the LLM-as-a-judge method has five ordinal bias categories, which are binarized after applying a threshold (scores higher than 2 reflect the presence of bias). Similarly, the output of other methods is also binarized to describe whether or not the thoughts are biased. The binarized scores of each method are then compared with the ground truth to compute the F1-scores, which reflect the performance of each method.”
>
> >How is the metric in 4.5 Natural language inference different from 4.6 Bias Reasoning Analysis using Information Norms? Is the only difference the model that were applied?
>
> In Section 4.5, we frame bias detection as a NLI task: the model’s thought is the premise, and a hypothesis tests whether it implies a biased or unbiased view. Bias is flagged if the model predicts entailment for the hypothesis (using BART and mT5).
>
> In Section 4.6, we take a probabilistic approach. BRAIN compares the probability distributions over answer choices generated from the [Question; Context] and [Question; Thought] inputs, using the Jensen-Shannon divergence as a metric. A high divergence score indicates a mismatch between the thought and the answer, suggesting bias when the thought aligns with a biased output or contradicts an unbiased one.
>
> >How are the "unbiased" thoughts obtained? Maybe the authors would want to revise the wording to make the meaning of "biased" or "unbiased" more consistent with the remainder of the paper.
>
> We agree with the reviewer that the terminology is confusing. Therefore, we will add this sentence to Section 1: “Throughout the paper, we refer to unbiased and fair thoughts interchangeably, which refers to the thoughts that do not arrive at conclusions based on race, religion, sexual orientation, nationality, gender identity, socio-economic status, age, disability, and physical appearance”.

---

> > ### Author Response · Authors · 2025-06-10
> > **Following up**
> >
> > Dear Reviewer bsru,
> >
> > Thanks for dedicating your time and effort to review our work. Your comments improved the clarity and soundness of our paper. Given that the discussion period is closing, we kindly ask if our responses have clarified your concerns. If not, we will be happy to provide further clarifications.

---

> > > ### Comment · Reviewer_bsru · 2025-06-10
> > >
> > > I imagine the paper will be of greater value to the community based on the proposed phrasing changes alone. Thanks for the clarifications.

---

> > > ### Comment · Reviewer_bsru · 2025-06-10
> > >
> > > Quick note regarding the qualitative results- these are very helpful. Thanks for sharing them. However, please do make sure that the "Thought" is produced before the "Answer", as the autoregressive LLMs can't go back and edit their response- only justifying their existing output- after producing the answer. (This point does not apply to models like gemini-diffusion)

---

> > > > ### Author Response · Authors · 2025-06-10
> > > > **Thanks for your engagement**
> > > >
> > > > We thank the reviewer for the important remarks and thoughtful engagement in the discussion. We agree that the thoughts in the qualitative analysis of autoregressive LLMs should be shown before the final answer. Therefore, we will edit our figures accordingly.

---

> > ### Author Response · Authors · 2025-06-10
> > **Updated figures**
> >
> > Thanks again for you important comments. We have updated the figures for the qualitative analysis as follows:
> >
> > https://drive.google.com/file/d/1GpjH82KM3XaJ9C7vQR0EbDJ6ZQ_vhinJ/view?usp=sharing
> >
> > https://drive.google.com/file/d/1QNWbCO1MY5UPlgIZxPH4wzCLykWYpj7D/view?usp=sharing

---

### Official Review · Reviewer_kJ6k · 2025-05-13

**Rating:** 6
**Confidence:** 4
**Ethics Flag:** 1

**Summary:**

This paper explores the research question on whether biased language models have biased thoughts. The paper studies 5 large language models using chain-of-thought method to see if bias in thinking steps connect with bias in final output. The result show that bias in thoughts not strongly related to bias in answers (correlation less than 0.6), unlike humans whose biased decisions usually come from biased thinking. The paper also reports that adding neutral thoughts to prompts can make models give less biased responses.

**Questions To Authors:**

* Lines 220: "We use BART Lewis et al. (2019) and mT5 Xue et al. (2021)". Brackets missing for the citations in this line.
  * This is also the case for Section 2.3, which makes it slightly harder to read.

###  Questions
* Lines 157-158: "we use a Llama model (70B)" but which one is used in the paper? Llama 3 70B, Llama 3.1-70B, or Llama 3.3 70B?
* Appendix E.6: Are these decoding hyperparameters tuned? Asking this since setting to < 1 temperature simply encourages non-diversity which results in more bias in the generation outputs.

**Reasons To Accept:**

* The main research question of the paper is simple but interesting that has not been deeply analyzed in other papers to the best of my knowledge.
* Good reproducibility since detailed hyperparameter setup is reported.
* Experimental results on 6 different bias measurements with clear differences between LLM judges & BRAIN vs. other approaches.

**Reasons To Reject:**

* The general takeaways are not super clear especially given that the correlation of 0.6 is moderate and largely differs across models and fine-grained takeaways per model level are missing (Figure 4).
* The experimented models in the paper are relatively smaller scale (<=8B).
* Fine-grained analysis on reasonings is missing: e.g., one reasoning step may include bias while another step may not include bias. This is not discussed or measured in the paper.

---

> ### Author Response · Authors · 2025-06-03
> **Thanks for dedicating your time**
>
> **Thank you for your detailed review. We’re encouraged to know that you found the main research question interesting. We address your main points below:**
> >The general takeaways are not super clear especially given that the correlation of 0.6 is moderate and largely differs across models and fine-grained takeaways per model level are missing (Figure 4).
>
> We agree with the reviewer that correlation values <0.6 do not reflect a high positive correlation. However, we believe that studying the correlation between thoughts and output in the domain of fairness is meaningful, particularly in light of prior work [1,2], which has suggested that intermediate thoughts often correlate with better final performance. Our results show that, in the context of fairness, this alignment does not consistently hold. In other words,  biased reasoning does not necessarily imply a biased final response.
> Our observation is novel and important for future studies on alignment and interpretability in language models. Thank you for pointing out the missing analyses per model. We will add the model-level analysis in the camera-ready submission.
>
> >The experimented models in the paper are relatively smaller scale (<=8B).
>
> We agree with the reviewer that, with the exception of Llama 70B (for which we have a limited quota) for annotation, all our models have <=8B parameters due to our computational constraints.
>
> >Fine-grained analysis on reasonings is missing: e.g., one reasoning step may include bias while another step may not include bias. This is not discussed or measured in the paper.
>
> This is an important remark. To address this limitation, we created Appendix A.5 for qualitative analysis in our paper, with the following examples:
>
> https://drive.google.com/file/d/1XAoiep7QRYqe7hB5rqsVZuzQWoe_Mfds/view?usp=sharing
>
> https://drive.google.com/file/d/18MSOfpzoeBAxSLbj5I1dpxWOsi9cXvNf/view?usp=sharing
>
> >Lines 220: "We use BART Lewis et al. (2019) and mT5 Xue et al. (2021)". Brackets missing for the citations in this line. This is also the case for Section 2.3, which makes it slightly harder to read.
>
> This is an important comment. We fixed all the citations for the camera-ready version.
> >Lines 157-158: "we use a Llama model (70B)" but which one is used in the paper? Llama 3 70B, Llama 3.1-70B, or Llama 3.3 70B?
>
> We apologize for the confusion. The following sentence has been added to Section 5.4:
> “We employed LLaMA 3 70B Instruct variant (meta-llama/Meta-Llama-3-70B-Instruct) to obtain the ground truth values for the presence of bias in the thoughts.”
> >Appendix E.6: Are these decoding hyperparameters tuned? Asking this since setting to < 1 temperature simply encourages non-diversity which results in more bias in the generation outputs.
>
> We thank the reviewer for raising this point. For generating intermediate thoughts and final answers, we used the default Hugging Face decoding parameters: temperature = 0.7 and top-k = 50. These values were selected to balance diversity and output format consistency. For the thought generation and ground truth predictions using Llama 70B (to generate 0/1 answers), we used a temperature of 0.01. While higher temperatures may increase response diversity, based on qualitative analysis of a few samples, we found that they often led to deviations from the expected output format specified in the prompt.
>
>
> [1] Wei, Jason, et al. "Chain-of-thought prompting elicits reasoning in large language models." Advances in neural information processing systems 35 (2022): 24824-24837.
>
> [2] Yee, Evelyn, et al. "Dissociation of faithful and unfaithful reasoning in llms." arXiv preprint arXiv:2405.15092 (2024).

---

### Official Review · Reviewer_d6Ma · 2025-05-14

**Rating:** 7
**Confidence:** 4
**Ethics Flag:** 1

**Summary:**

This paper analyses biases in reasoning steps (in chain-of-thought) and how it affects the answers in a QA setting.

**Questions To Authors:**

- Please fix your citations, you are consistently using textual citations (\citet{...}) instead of the more appropriate \citep{..} or \cite.
- As a writing suggestion, I would perhaps merge Subsection 2.1 and Section 3.
- I am confused by the information in L158 and L164. They seem to be geared towards the same task, but with different models. Which one was actually used? I also don't think the distilled R1 8B model is the best option, so it would be cool if you could re-run this?
- I do not fully understand the differences between Experiment 1 and Experiment 2. Both use the label of the BBQ dataset and measure how well the classifications of the reasoning steps matches this label?

**Reasons To Accept:**

The paper is timely and well-executed. The evaluation builds upon relevant related work, investigates the reliability of their measures and evaluates multiple models.

There are interesting findings regarding biased reasoning and the additional experiment of including unbiased reasoning seems to reduce some biases.

**Reasons To Reject:**

While the paper is interesting and seems scientifically sound enough, I had some trouble understanding the different experiments and their exact setup. The writing could be improved. In addition, a figure describing the evaluation setting more in-depth (Figure 2 is a good start) could also be very helpful. Overall, the paper is trying to do a lot and I believe it could benefit from some streamlining.

Additionally, the models studies are also not reasoning models. Up until Section 5.4 I was expecting an analysis of _reasoning_ models instead of chain-of-thought traces. Perhaps some additional experiments on these kinds of models could be useful? There seems to be some mentions of a reasoning model (distilled R1 8B), but it is unclear in which setting this was used.

Finally, some clearer positional statement (e.g. a Bias statement or broadening the ethics statement) would be nice. It is a bit unclear which biases are exactly investigated, as the reference to BBQ is doing a lot of hte heavy lifting here. Also I assume this is for English, please make that explicit as well? Some relevant work here [1]

[1] https://aclanthology.org/2024.emnlp-main.1207/

---

> ### Author Response · Authors · 2025-06-03
> **Thank you for your time and effort**
>
> **Thank you for your detailed review and suggestions for improving this study. We’re glad that you found our work scientifically sound, timely, and well-executed. We address your main points below:**
> >I had some trouble understanding the different experiments and their exact setup. The writing could be improved.
> >I do not fully understand the differences between Experiment 1 and Experiment 2
>
> Thank you for your valuable feedback. We worked on improving the clarity of our writing and will add more detailed figures to better illustrate our experimental design. We conducted four experiments:
>
> **Experiment 1** focused on how to quantify bias in the thoughts. We evaluated six different methods. The input is the model's thoughts, and the output is whether or not they are biased.
>
> **Experiment 2** focused on answering our main question: Do models with biased outputs have biased thoughts? We answered this question by measuring the correlation between bias in the output and in the thoughts. The answer is: No.
>
> **Experiment 3** answered a side question: Does answering in a step-by-step way increase/decrease bias in the output? The answer is: No. The input is the question, and the output is the model's thoughts and answer.
>
> **Experiment 4** answered another side question: Does adding unbiased thoughts to the question make the answer fairer? The answer is: Yes. The input is the question with unbiased thoughts, and the output is the answer.
>
>
> > a figure describing the evaluation setting more in-depth
>
> Thanks for the suggestion. To address this, we created Appendix A.5 for qualitative analysis in our paper, with the following examples:
>
> https://drive.google.com/file/d/1XAoiep7QRYqe7hB5rqsVZuzQWoe_Mfds/view?usp=sharing
>
> https://drive.google.com/file/d/18MSOfpzoeBAxSLbj5I1dpxWOsi9cXvNf/view?usp=sharing
>
> >Additionally, the models studies are also not reasoning models.
>
> Thank you for your feedback. We analyze LLM outputs and their corresponding thoughts, rather than evaluating reasoning processes. In other words, we do not measure how the model arrives at an answer, but rather whether the generated thought and final answer are biased, and whether there is a correlation between the two.
> The distilled R1 8B model is used in one of our six evaluation methods: the LLM-as-a-judge. In this setup, the distilled R1 8B model is tasked with analyzing and quantifying bias in the generated thoughts.
>
> >Finally, some clearer positional statement (e.g. a Bias statement or broadening the ethics statement) would be nice. Also I assume this is for English, please make that explicit as well? Some relevant work here [1] https://aclanthology.org/2024.emnlp-main.1207/
>
> Thanks for your thoughtful observation and for pointing out that our study is limited to the English language. We will revise our ethics statement accordingly, by adding the following sentence: “Our study covers 11 social biases, namely: age, disability status, gender identity, nationality, physical appearance, race/ethnicity, race/socioeconomic status (SES), race/gender intersection, religion, sexual orientation, Socioeconomic status (SES). This study is limited to the English language.”
>
> >Please fix your citations, you are consistently using textual citations (\citet{...}) instead of the more appropriate \citep{..} or \cite.
>
> Thanks for pointing it out. We fixed all the textual citations.
> >As a writing suggestion, I would perhaps merge Subsection 2.1 and Section 3.
>
> We appreciate the reviewer’s helpful suggestion to improve our writing. We will merge Subsection 2.1 and Section 3 in the camera-ready version.
>
> >I am confused by the information in L158 and L164.
>
> We agree with the reviewer that these lines need more clarification. To clarify, we employed LLaMA 70B as a strong open-source model to obtain the ground truth labels. In this setup, the model was prompted to return a binary decision (0 or 1) indicating whether the "thinking steps" of another model are biased.
> Separately, we used the DeepSeek R1 8B distilled model as the LLM-as-a-judge baseline. This model was prompted to assign a score between 0 and 5, reflecting the degree of perceived bias in the thoughts.
> We will add the following sentence before Subsection 4.1 to clarify this confusion:
> “It is important to distinguish between the LLM-as-a-judge baseline and the LLaMA 70B model. LLaMA 70B is used as an annotator to all the baselines, including the LLM-as-a-judge.”

---

> > ### Comment · Reviewer_d6Ma · 2025-06-04
> >
> > While I won't adjust my score, I am happy with the responses and hope the authors do the promised work for the camera-ready.

---

> > > ### Author Response · Authors · 2025-06-04
> > > **Thank you**
> > >
> > > We thank the reviewer for the time and effort dedicated to provide us with a detailed review. We will do all the promised suggestions in the camera-ready version of the paper to improve our work. In case there are any other comments/questions, we will be happy to discuss them here.

---

### Decision · Program_Chairs · 2025-07-08

**Decision:**

Accept

**Comment:**

This paper investigates whether biased language models show biased reasoning in their chain-of-thought outputs, finding that bias in intermediate reasoning steps only has a moderate correlation (<0.6) with the final answer bias. Reviewers appreciate the timely research question and the fact that experiments consider multiple bias detection methods across several models.

The discussion phase was productive, and authors addressed some key reviewer concerns. Reviewers bsru and iGTF both raised their scores after engaging with the authors. The finding that biased thoughts don't strongly predict biased output may be surprising to some (it challenges some assumptions in prior work) and can have implications for mechinterp and alignment/safety research. The experiments on injecting unbiased thoughts as a bias mitigation strategy are interesting and novel, and can open some new research directions.